# Effects of Acupuncture-Related Therapies in the Rehabilitation of Patients with Post-Stroke Aphasia—A Network Meta-Analysis of Randomized Controlled Trials

**DOI:** 10.3390/brainsci12101282

**Published:** 2022-09-23

**Authors:** Pengpeng Liang, Yufei Li, Yanan Feng, Guoliang Yin, Suwen Chen, Xiangyi Liu, Fengxia Zhang

**Affiliations:** 1The First Clinical Medical College, Qianfoshan Campus, Shandong University of Traditional Chinese Medicine, 16369 Jingshi Rd., Jinan 250000, China; 2Department of Neurology, Affiliated Hospital of Shandong University of Traditional Chinese Medicine, 16369 Jingshi Rd., Jinan 250011, China

**Keywords:** acupuncture, post-stroke aphasia, randomized controlled trials, network meta-analysis

## Abstract

Objective: The purpose of this study was to evaluate the rehabilitation effects of four common interventions (BA: body acupuncture, SA: scalp acupuncture, TA: tongue acupuncture, SLT: speech and language training) used singly or in combination with language function in patients with post-stroke aphasia (PSA). Design: We systematically searched PubMed, EMBASE, Cochrane Library, Ovid, Web of Science, CNKI, VIP, and Wanfang from inception to 4 April 2022. Only randomized controlled trials that met the eligibility criteria were included. The risk of bias of studies included was assessed using the RoB-2 tool. The effects of different interventions for PSA patients were analyzed and ranked according to the surface under the cumulative ranking (SUCRA) analysis. Results: A total of 69 RCTs were included, including 5097 total participants. According to the results of the SUCRA curves, TA ranked highest in improving overall efficacy (SUCRA = 86%) and oral expression score (SUCRA = 86%). BA + TA ranked highest in increasing the comprehension score (SUCRA = 74.9%). BA + SA ranked highest in improving aphasia patients’ repetition (SUCRA = 89.2%) and denomination scores (SUCRA = 93%). Conclusions: Results of our network meta-analysis and SUCRA ranking showed that tongue acupuncture, body acupuncture + tongue acupuncture, and body acupuncture + scalp acupuncture seem to offer better advantages than other interventions for improving the language function in PSA patients. Moreover, it is noteworthy that our results are limited to the Chinese population, since all eligible studies are from China. Future well-designed studies with larger sample sizes and more ethnic groups are required to further verify these findings.

## 1. Introduction

Today, stroke is the third most common cause of death and the leading cause of permanent adult disability in the world. It is estimated that there are about 13 million new stroke cases worldwide each year, with over 5.5 million people suffering from strokes in China [1]. Approximately 21–38% of stroke survivors suffer post-stroke aphasia (PSA) [2]. PSA, a common language disorder after stroke, is characterized by impairment of partial or all language functions, including oral expression, comprehension, and reading and writing [3]. It is mainly caused by strokes in the left hemisphere and severe trauma or neurodegenerative illness affecting the brain’s language network pathways, primarily Broca and Wernicke areas [4]. Speech is a crucial instrument for thinking and communicating in human social activities. Aphasia not only impairs the ability of patients to participate in social activities and career development and reduces their quality of life, but also causes various psychological symptoms such as loneliness, anxiety, and depression, which impose a substantial emotional and financial burden on families and society.

Current treatments for PSA include speech language training, medication, Virtual Reality (VR), and transcranial direct current stimulation (tDCS) [5]. Despite all the benefits of clinical therapies, these approaches are still faced with challenges and limitations to their clinical utility. Speech and language training is an effective strategy to improve language dysfunction, considered a gold treatment for PSA [6]. Meanwhile, a series of studies [7,8,9] have shown that speech language training significantly improves the reconstruction of language function in patients with aphasia after stroke. However, in China, the speech therapy industry is in its infancy, and the trained and certified speech therapists are unevenly distributed throughout the country, mainly concentrated in developed cities [10]. In addition, speech therapists must construct tailored rehabilitation plans for patients, and factors such as long-term training and high expenses are challenging for patients to comply with them. Recently, some drugs that improve neural plasticity have been tried for post-stroke aphasia [11]. Despite the theory that pharmacotherapy modulates the activity of neurotransmitter systems, which is a promising strategy for restoring language and communication deficits in PSA patients [12,13,14], the results of many RCTs of pharmacological interventions for stroke aphasia are inconsistent [15,16,17,18,19,20,21]. More clinical trial data and animal studies are required to further explore the molecular mechanisms and safety of drugs. Therefore, exploring a complementary and alternative method to treat PSA is urgently necessary.

Acupuncture, a significant component of TCM treatment, has been widely used in various diseases, including neurological dysfunctions [22,23]. More studies have shown that acupuncture can alleviate participants’ functional symptoms and improve their language ability after stroke. A meta-analysis including 28 RCTs involving 1747 patients showed that acupuncture effectively improves functional communication in PSA [24]. Similarly, the meta-analysis conducted by Zhang also reveals that acupuncture used alone or in combination with SLT yielded therapeutic benefits in the rehabilitation of PSA patients [25]. Still, more prospective randomized clinical trials are needed to evaluate and confirm their findings. Furthermore, an fMRI study showed that the therapeutic benefits of acupuncture in PSA patients might be related to the activation and functional connectivity of language-related brain areas on their lesion side [26]. However, to date, the efficacy and mechanism of acupuncture in treating aphasia are still unclear. Furthermore, which acupuncture therapy is more clinically beneficial for treating PSA remains to be confirmed. Network meta-analysis (NMA), as an essential tool for evidence-based medicine, can solve this situation by calculating the comparative effects of multiple treatments on a disease and ranking treatments from direct and indirect evidence, whereas a conventional meta-analysis can only achieve pairwise direct comparison of intervention measures in clinical trials. [27,28,29,30,31]. In this study, we conducted a network meta-analysis to systematically compare the effects of a range of interventions of three common acupuncture and speech language training methods used singly or in combination in treating PSA patients.

## 2. Materials and Methods

### 2.1. Search Strategy

We systematically searched PubMed, Embase, Web of Science, Ovid, the Cochrane Library, and three Chinese databases, CNKI, Wanfang and VIP, for publications on the effectiveness of acupuncture-related therapies for post-stroke aphasia from these databases’ inception to April 2022. According to the PICOS principle (Population, Intervention, Comparison, Outcome, Study design), we developed the search strategy of a combination of subject terms and free terms. Our retrieval scheme included various medical subject headings and free text words related to stroke, cerebrovascular disease, aphasia, and acupuncture to obtain extensive literature for further analyses. Taking Pubmed as an example, our search strategy is provided in the Appendix A.

### 2.2. Inclusion Criteria

The search results were evaluated according to guidelines provided in the Preferred Reporting Items for Systematic Reviews and Meta-Analyses for Network Meta-Analyses (PRISMA-NMA) checklist [32], and the protocol was registered in INPLAYSY (registration number INPLASY202290024) (https://inplasy.com/inplasy−2022−9−0024/ (accessed on 18 September 2022)). Criteria for inclusion are as follows: (1) Only randomized controlled trials (RCTs) were included in this study. (2) Participants enrolled in the study were diagnosed using an aphasia function scale and neuroimaging techniques, CT or MRI, with no restrictions on type, gender, or age. (3) Three kinds of acupuncture (body acupuncture, scalp acupuncture, tongue acupuncture), alone or combined with speech rehabilitation training, were employed as an intervention method. (4) The control group received either language rehabilitation or a kind of acupuncture selected by inclusion criteria. (5) Studies were published in English or Chinese. (6) The standardized Aphasia Battery of Chinese (ABC) scale was used to reflect whether acupuncture improves language function in PSA patients objectively. The ABC scale is the Chinese standardized adaptation of the Western Aphasia Battery according to the characteristics of Chinese culture and educational attainment [33,34]. The total effective rate was the primary indicator. With reference to the *Guiding Principles for Clinical Study of New Chinese Medicines* [35], the comprehensive curative effect criteria are as follows. The comprehensive curative effect = (Post-treatment ABC general score-Pre-treatment ABC general score) ÷ Pre-treatment ABC general score × 100%. (i) Substantially recovered: the comprehensive curative effect ≥95.0%. (ii) Markedly effective: the comprehensive curative effect ≥70.0% but <95.0%. (iii) Effective: the comprehensive curative effect ≥30.0% but <70.0%. (iv) Invalid: the comprehensive curative effect <30.0%. The overall response rate = (substantially recovered + markedly effective + effective)/overall subjects × 100%. Scores in comprehension, verbal expression, repetition, and denomination functions were derived from the ABC scales of each study as a secondary indicator and collated in the data extraction sheets.

### 2.3. Exclusion Criteria

Criteria for exclusion are as follows: (1) with other neurological diseases; (2) without precise diagnosis or inconsistent with inclusion diagnosis; (3) without outcome indicators or inconsistent with study indicators; (4) the intervention measures are inconsistent with the inclusion criteria; (5) repetitively published studies or without complete data in the study even after contacting the authors; (6) systematic review, meta-analysis, theoretical research, expert commentary, animal experiment, conference report, economic analysis, or case report.

### 2.4. Study Selection

EndNote (Version *X9*, Clarivate, Philadelphia, PA, USA) was used to process search records. Two reviewers (Y. Li and G. Yin) independently screened the title and abstract using the predetermined inclusion criteria. After that, the full text was further screened to exclude the literature which did not meet the inclusion criteria. Finally, the remaining literature was identified for inclusion by both authors. During this process, any discrepancies were discussed and resolved by a third author (Y. Feng).

### 2.5. Data Extraction

Data extraction was performed using an electronic form. The data details were planned to extract the author, year of publication, specific information about the treatment and control groups, and outcome indicators. Any disagreement between the two reviewers (P. Liang and G. Yin) was judged by the third reviewer (Y. Feng).

### 2.6. Assessment of the Risk of Bias

Two reviewers (X. Liu and S. Chen) independently used the Cochrane RoB−2 tool to assess the risk of bias in each study. The assessment includes evaluation in the following seven domains: random sequence generation, allocation concealment, blinding, incomplete outcome data, selective outcome reporting, and other possible biases. The studies were classified into low risk, high risk, and unclear risk. Discrepancies between the reviewers were solved through discussions with a third reviewer (Y. Feng) [36].

### 2.7. Statistical Analysis

In this study, odds ratios were used for dichotomous outcomes, and continuous variables in the study were reported as mean difference (MD = the difference in means between the treatment and control groups and calculated using the same scale) or standardized mean difference (SMD = mean difference in outcome between groups/standard deviation of outcome between subjects, used to combine data when trials have different scales) with estimated 95% confidence intervals (95% CI). To take into account differences between studies, we synthesized data using a random-effects model [37].

We performed a random-effects network meta-analysis within a frequentist framework using Stata/SE (version 15.0), and the results are presented with the mvmeta package [38]. A frequentist framework allowed us use *p*-values to make statistical comparisons [39]. Meanwhile, it was simpler in model specification and more useful for doctors to solve clinical practical problems compared with Bayesian framework [40]. Inconsistency in the network meta-analysis involved the design-by-treatment inconsistency model and the node-splitting approach [41]. The network evidence plots were drawn to summarize the geometry of the network of evidence. In the network evidence plot, each dot represents a different intervention, and the lines represent head-to-head comparisons. The size of each node and the width of the connecting lines are proportional to the number of studies. Simultaneously, the P rank score of the surface under the cumulative ranking curve (SURCA) was calculated to compare the rank of effect estimation for each treatment. SUCRA was expressed as a percentage (range, 0–100%): a larger value indicated a better intervention grade in sort results, and when the SUCRA value was close to 100%, it indicated that the treatment was the best one [42]. Efficacy of different treatments was displayed in a league table. Treatments were ranked from best to worst along the leading diagonal. Data presented are the ORs and MD (with 95% CI) in the column-defining treatment compared to the row-defining treatment for total clinical effective rate and ABC scale. OR above 1 indicated higher efficacy in improving language function, and MD above 0 favored the treatment in column in improving ABC scores. Effects were considered significant when OR and MD did not contain 1 and 0. Significant results are showed in bold. In addition, a “comparison-correction” funnel plot was used to assess the publication bias of the included studies [43]. Notably, the present results are based on the pooled data from each study included. Clinicians should interpret them cautiously and make reasonable clinical decisions according to the specific clinical situation [44].

## 3. Results

### 3.1. Results of Study Identification and Selection

A total of 1803 studies were included via the original search terms, and 526 articles were excluded due to duplication. After screening for title and abstract, 462 articles that failed the inclusion criteria were deleted, and after reading the whole text, 746 studies (84 non-RCT studies, 307 studies that did not meet the diagnosis of post-stroke aphasia, 266 studies whose interventions and controls were inconsistent with the criteria, 51 studies with duplicate, missing, or incalculable data, and 38 studies without targeted outcome indicators) were again removed. Finally, 69 studies were included into the risk of bias assessment and NMA. The process is presented in Figure 1.

### 3.2. Characteristics of the Included Studies

We finally included 69 RCTs, with 2590 participants in the intervention group and 2507 participants in the control group. In the included 69 studies, 67 were published in Chinese and 2 were published in English. The publication time of the studies was distributed from 1999 to 2022. Six studies reported BA [45,46,47,48,49,50], six reported TA [51,52,53,54,55,56], four reported BA + TA [57,58,59,60], nine studies reported SLT + SA [61,62,63,64,65,66,67,68,69], five studies reported SLT + SA + TA [70,71,72,73,74], fifteen reported SLT + BA [75,76,77,78,79,80,81,82,83,84,85,86,87,88,89], nine reported SLT + BA + SA [90,91,92,93,94,95,96,97,98], three reported SLT + BA + SA + TA [99,100,101], three studies reported SLT + BA + TA [102,103,104], and six studies reported SLT + TA [105,106,107,108,109,110]. The remaining three papers reported BA + SA [111], BA + SA + TA [112], and SA [113], respectively. The primary outcome of the included studies was the total effective rate; the secondary outcomes were the ABC scores (including comprehension, oral expression, repetition, denomination) and adverse effects. The characteristics of the included studies are shown in Table 1, and the details of interventions are shown in Appendix A.

### 3.3. Quality Assessment of the Included Studies

We assessed the risk of bias for each study using the Cochrane RoB−2 tool (Appendix A). In 69 included trials, 40 trials were at low risk of selection bias related to sequence generation because they used appropriate methods for random sequence generation, such as computer randomization or a random numbers table. Of the included trials, 26 that only mentioned random were evaluated as unclear risk. The remaining three trials used the order of consultation to generate random sequence and thus were rated as high risk of bias. We assessed all trials as unclear bias due to insufficiently reported allocation concealment methods. For blinding of subjects and outcome assessment, we evaluated almost all articles as high or unclear risk of bias. Only one article, Wang 2011, was accessed as at low risk of bias in blinding outcomes. All included trials carried a high risk for the blinding of subjects and outcome assessment. Regarding the risk of incomplete outcome data, 69 articles included in the current study showed an overall low risk of bias. Except for He A 2014, which was at high risk of bias, the remaining 68 studies were at low risk of selective reporting. Only one paper, Wei 2021, which reported no conflicts of interest, was rated as low risk of other bias. The remaining studies were considered to have an unclear risk of bias. Details of the risk of bias are presented in Appendix A.

### 3.4. Results of Network Meta-Analysis

#### 3.4.1. Evidence Network Diagram

A total of 63 studies with 4728 participants reported the total effective rate, involving 14 interventions: BA, BA + SA, BA + SA + TA, BA + TA, SA, SLT, SLT + BA, SLT + BA + SA, SLT + BA + SA + TA, SLT + BA + TA, SLT + SA, SLT + SA + TA, SLT + TA, and TA [45,46,47,48,49,50,52,53,54,55,56,57,58,59,60,62,63,64,65,66,67,68,69,70,71,72,73,74,75,76,77,78,79,80,83,84,85,86,87,88,89,90,91,92,93,94,95,96,97,98,99,100,101,102,103,104,105,106,108,110,111,112,113]. A total of 22 studies with 1573 participants reported the comprehension score, including 14 interventions: BA, BA + SA, BA + SA + TA, BA + TA, SA, SLT, SLT + BA, SLT + BA + SA, SLT + BA + SA + TA, SLT + BA + TA, SLT + SA, SLT + SA + TA, SLT + TA, and TA [45,48,57,58,59,65,66,71,73,75,81,82,85,94,95,100,103,107,109,111,112,113]. A total of 21 studies with 1427 participants reported the oral expression score, including 14 interventions: BA, BA + SA, BA + SA + TA, BA + TA, SA, SLT, SLT + BA, SLT + BA + SA + TA, SLT + BA + TA, SLT + SA, SLT + SA + BA, SLT + SA + TA, SLT + TA, and TA [45,48,51,57,58,59,61,65,66,73,81,85,94,95,100,103,107,109,111,112,113]. A total of 22 studies with 1557 participants reported the repetition score, including 13 interventions: BA, BA + SA, BA + SA + TA, BA + TA, SA, SLT, SLT + BA, SLT + BA + SA + TA, SLT + BA + TA, SLT + SA, SLT + SA + TA, SLT + TA, and TA [45,48,51,57,58,59,61,65,71,73,75,81,82,85,95,100,103,107,109,111,112,113]. A total of 21 studies with 1497 participants reported the denomination score, including 13 interventions: BA, BA + SA, BA + SA + TA, BA + TA, SA, SLT, SLT + BA, SLT + BA + SA, SLT + BA + SA + TA, SLT + BA + TA, SLT + SA, SLT + SA + TA, and SLT + TA [45,48,57,58,59,61,65,71,73,75,81,82,85,95,100,103,107,109,111,112,113]. The details of the NMA figures were shown in Figure 2, Figure 3, Figure 4, Figure 5 and Figure 6A.

#### 3.4.2. Results of Total Effective Rate

A total of 63 studies were aggregated to compare the effect of each intervention on the total clinical effective rate [45,46,47,48,49,50,52,53,54,55,56,57,58,59,60,62,63,64,65,66,67,68,69,70,71,72,73,74,75,76,77,78,79,80,83,84,85,86,87,88,89,90,91,92,93,94,95,96,97,98,99,100,101,102,103,104,105,106,108,110,111,112,113]. All *p*-values of the two methods for accessing inconsistency (Appendix A) are greater than 0.05, indicating excellent consistency and stability of studies.

Our NMA results showed that all the intervention measures were statistically significant, indicating that acupuncture has a significant effect on improving language function. The details are shown in Table 2. The SUCRA rankings of all treatments for the clinical effective rate are shown in Figure 2B. According to the results of SUCRA analyses, TA (SUCRA, 86%) was likely to be the most effective intervention to improve post-stroke aphasia, followed by BA + SA (SUCRA, 75.7%), BA + TA (SUCRA, 73%), SLT + BA + SA (SUCRA, 70.4%), SLT + SA + TA (SUCRA, 64.3%), SLT + BA + TA (SUCRA, 60.3%), SLT + BA (SUCRA, 57.6%), SLT + BA + SA + TA (SUCRA, 51%), BA + SA + TA (SUCRA, 50%), SLT + TA (SUCRA, 35.4%), SLT + SA (SUCRA, 34.3%), BA (SUCRA, 22.3%), SA (SUCRA, 17.8%), and SLT (SUCRA, 1.8%).

**Figure 2 brainsci-12-01282-f002:**
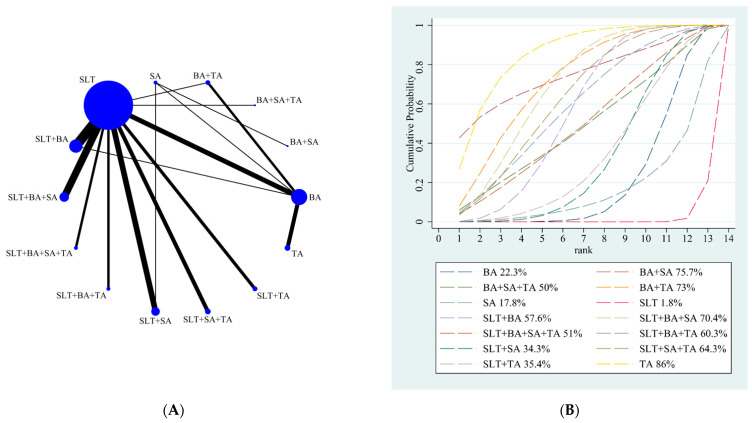
(**A**) NMA figure for total effective rate. (**B**). SUCRA plot for total effective rate.

#### 3.4.3. Results of the Comprehension Score

A total of 22 studies were aggregated to compare the effect of each intervention on the comprehension score [45,48,57,58,59,65,66,71,73,75,81,82,85,94,95,100,103,107,109,111,112,113]. All *p*-values of the two methods for accessing inconsistency (Appendix A) are greater than 0.05, indicating excellent consistency and stability of studies.

Our NMA results showed that BA (21.02; 95% CI, 0.12 to 42.32) and BA + SLT (15.09; 95% CI, 0.21 to 29.96) were better than SLT. The other different treatments showed no significant difference. The details are shown in Table 3. According to the results of SUCRA analyses, BA + TA (SUCRA, 74.9%) had the highest probability of being the best treatment for increasing comprehension score, followed by BA + SA (SUCRA, 72.3%), BA (SUCRA, 67.1%), TA (SUCRA, 65.4%), SLT + BA (SUCRA, 60.9%), SLT + SA + TA (SUCRA, 56.2%), SA (SUCRA, 54.7%), SLT + TA (SUCRA, 48%), SLT + BA + TA (SUCRA, 47.6%), SLT + BA + SA (SUCRA, 42.2%), SLT + SA (SUCRA, 35.9%), BA + SA + TA (SUCRA, 31.2%), SLT + BA + SA + TA (SUCRA, 22.3%), and SLT (SUCRA, 21.2%). The SUCRA rankings of all treatments for the comprehension score are shown in Figure 3B.

**Figure 3 brainsci-12-01282-f003:**
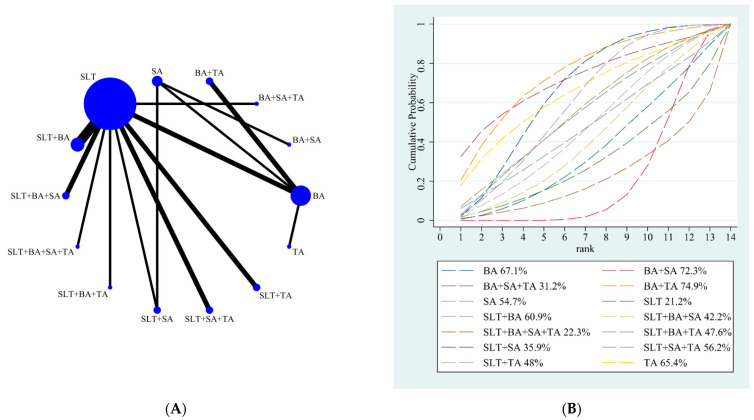
(**A**) NMA figure for the comprehension score. (**B**) SUCRA plot for the comprehension score.

#### 3.4.4. Results of the Oral Expression Score

A total of 21 studies were aggregated to compare the effect of each intervention on the oral expression score [45,48,51,57,58,59,61,65,66,73,81,85,94,95,100,103,107,109,111,112,113]. All *p*-values of the two methods for accessing inconsistency (Appendix A) are greater than 0.05, indicating excellent consistency and stability of studies.

Our NMA results showed only BA + TA (18.08; 95% CI, 1.46 to 34.69) was better than SLT. The other different treatments showed no significant difference. The details are shown in Table 4. According to the results of SUCRA analyses, the top-ranked intervention for boosting oral expressing function was TA, followed by BA + TA (SUCRA, 83.2%), BA + SA (SUCRA, 68.1%), BA (SUCRA, 57.3%), SLT + SA + BA (SUCRA, 55.3%), SLT + BA (SUCRA, 52.8%), SLT + TA (SUCRA, 48.8%), SLT + SA + TA (SUCRA, 45.6%), SLT + SA (SUCRA, 43%), SA (SUCRA, 40.7%), SLT + BA + TA (SUCRA, 33.3%), BA + SA + TA (SUCRA, 32%), SLT + BA + SA + TA (SUCRA, 31%), and SLT (SUCRA, 22.9%). The SUCRA rankings of all treatments for oral expression are shown in Figure 4B.

**Figure 4 brainsci-12-01282-f004:**
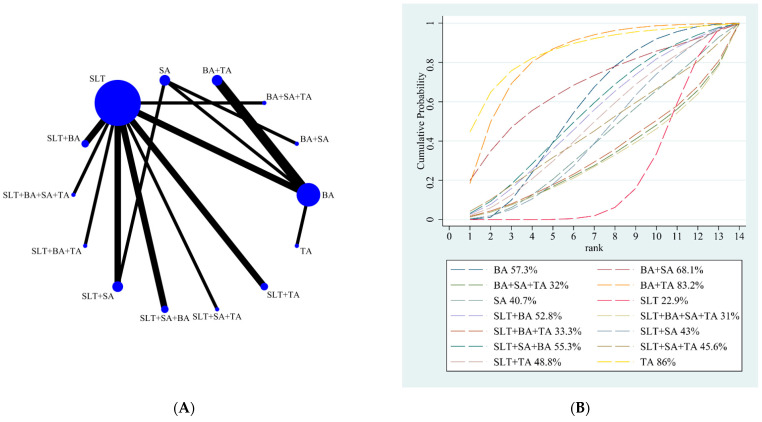
(**A**) NMA figure for the oral expression score. (**B**) SUCRA plot for the oral expression score.

#### 3.4.5. Results of the Repetition Score

A total of 22 studies were aggregated to compare the effect of each intervention on the repetition score [45,48,51,57,58,59,61,65,71,73,75,81,82,85,95,100,103,107,109,111,112,113]. All *p*-values of the two methods for accessing inconsistency (Appendix A) are greater than 0.05, indicating excellent consistency and stability of studies.

Our NMA results showed that BA + SA (25.55; 95% CI, 3.59 to 47.52), SLT + TA (21.50; 95% CI, 6.74 to 36.26), BA + TA (19.98; 95% CI, 8.88 to 31.08), BA (12.56; 95% CI, 3.53 to 21.58), SLT + SA (11.58; 95% CI, 3.59 to 19.57), SLT + BA (10.83; 95% CI, 5.19 to 16.47), and SLT + BA + SA + TA (8.15; 95% CI, 0.35 to 15.95) were better than SLT. BA + SA (24.50; 95% CI, 0.07 to 48.93), SLT + TA (20.45; 95% CI, 2.21 to 38.68), and BA + TA (18.93; 95% CI, 3.51 to 34.35) were better than BA + SA + TA. SLT + TA (19.47; 95% CI, 2.23 to 36.70) and BA + TA (17.95; 95% CI, 3.73 to 32.17) were better than SLT + BA + TA. SLT + TA (20.55; 95% CI, 2.07 to 39.02) and BA + TA (19.03; 95% CI, 3.33 to 34.73) were better than SLT + BA + SA. BA + TA (7.42; 95% CI, 0.96 to 13.89) was better than BA. The details are shown in Table 5. According to the results of SUCRA analyses, BA + SA (SUCRA, 89.2%) was likely to be the most effective intervention to improve post-stroke aphasia, followed by TA (SUCRA, 85.8%), BA + TA (SUCRA, 84.5%), SA (SUCRA, 61.9%), BA (SUCRA, 59.7%), SLT + SA + TA (SUCRA, 59.5%), SLT + BA (SUCRA, 56.9%), SLT + BA + TA (SUCRA, 45.3%), SLT + TA (SUCRA, 38.2%), SLT + SA (SUCRA, 20.8%), BA + SA + TA (SUCRA, 18.8%), SLT + BA + SA + TA (SUCRA, 18.4%), and SLT (SUCRA, 11.1%). The SUCRA rankings of all treatments for the repetition score are shown in Figure 5B.

**Figure 5 brainsci-12-01282-f005:**
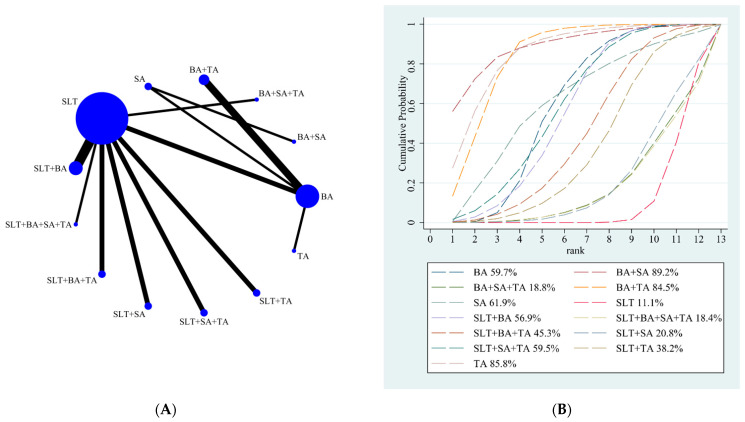
(**A**) NMA figure for the repetition score. (**B**) SUCRA plot for the repetition score.

#### 3.4.6. Results of the Denomination Score

A total of 21 studies were aggregated to compare the effect of each intervention on the denomination score [45,48,57,58,59,61,65,71,73,75,81,82,85,95,100,103,107,109,111,112,113]. All *p*-values of the two methods for accessing inconsistency (Appendix A) are greater than 0.05, indicating excellent consistency and stability of studies.

Our NMA results showed that BA + SA (29.47; 95% CI, 4.32 to 54.63), BA + TA (22.66; 95% CI, 7.31 to 38.02), and BA (17.52; 95% CI, 3.74 to 31.29) were better than SLT + SA. BA + SA (28.65; 95% CI, 5.29 to 52.01), BA + TA (21.83; 95% CI, 9.65 to 34.02), BA (16.69; 95% CI, 6.56 to 26.81), SLT + BA (9.82; 95% CI, 3.83 to 15.81), and SLT + SA + TA (9.01; 95% CI, 0.62 to 17.40) were better than SLT. BA + SA (28.44; 95% CI, 2.38 to 54.51), BA + TA (21.63; 95% CI, 4.83 to 38.43), and BA (16.49; 95% CI, 1.11 to 31.86) were better than SLT + BA + SA + TA. BA + SA (28.08; 95% CI, 2.02 to 54.15), BA + TA (21.27; 95% CI, 4.47 to 38.08), and BA (16.13; 95% CI, 0.75 to 31.50) were better than BA + SA + TA. BA + TA (15.49; 95% CI, 0.45 to 30.53) was better than SLT + TA. The details are shown in Table 6. According to the results of SUCRA analyses, BA + SA (SUCRA, 93%) was likely to be the most effective intervention to improve post-stroke aphasia, followed by BA + TA (SUCRA, 88.5%), BA (SUCRA, 75%), SA (SUCRA, 74.1%), SLT + BA (SUCRA, 58.2%), SLT + SA + TA (SUCRA, 54.4%), SLT + BA + SA (SUCRA, 48.4%), SLT + TA (SUCRA, 44%), SLT + BA + TA (SUCRA, 38.3%), BA + SA + TA (SUCRA, 22.4%), SLT + BA + SA + TA (SUCRA, 21.2%), SLT + SA (SUCRA, 16.3%), and SLT (SUCRA, 16.2%). The SUCRA rankings of all treatments for the clinical effective rate are shown in Figure 6B.

**Figure 6 brainsci-12-01282-f006:**
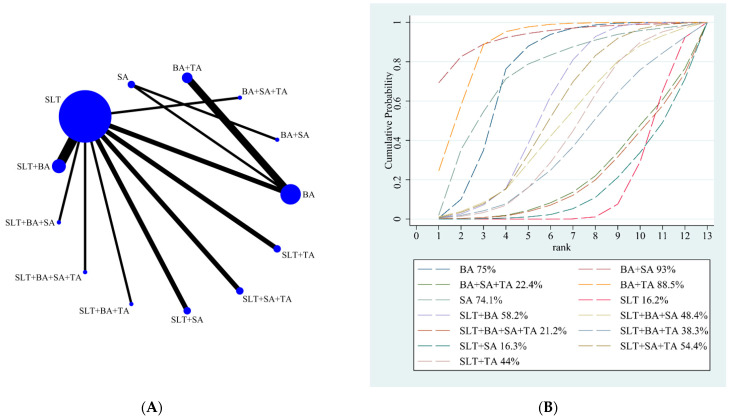
(**A**) NMA figure for the denomination score. (**B**) SUCRA plot for the denomination score.

### 3.5. Presence of Adverse Effects

None of the included studies reported adverse effects, so the network meta-analysis could not be further performed.

### 3.6. Publication Bias and Consistency Assessment

We constructed a comparison–correction funnel plot of the primary outcome of the clinical effective rate by Stata/SE 15.0 for evaluation. As shown in Figure 7, the funnel plot shows that the scattered points are symmetrically distributed within the funnel borders, suggesting limited publication bias.

## 4. Discussion

### 4.1. This Work

With the continuous progress of diagnostic techniques and medical treatments, the condition of most stroke patients has been effectively controlled in the acute and subacute phases but still leaves a series of sequelae. PSA is one of the most severe and persistent symptoms in cerebrovascular sequelae [114,115,116,117]. Losing language functions in PSA patients leads to a decline in quality of life and a massive psychological burden on patients and their families [118,119,120]. Over the past decades, many research teams have devoted substantial effort to studying PSA’s potential mechanisms and therapeutic approaches. It is generally accepted that PSA’s pathogenesis was related to functional impairment, hypoperfusion, and hypometabolism of cerebrocortical areas after stroke [121,122,123,124]. The clinical strategy for PSA focuses on improving specific language function deficits of PSA patients by using various approaches to enhance functional communication. However, these treatments show no significant differences in efficacy, and there is no universally recognized standard treatment scheme for PSA patients [125]. Their therapeutic effects are also affected by many factors, such as exposure, the level of medical service, dose, course of treatment, patient compliance, sample size, and so on. Therefore, given the unsatisfactory efficacy of current treatments, it is crucial to find alternative or complementary therapies. Compared with many rehabilitation therapies, acupuncture has the advantages of safety, simple operation, non-toxic side effects, and relatively low cost. It has more than 3500 years of medical practice history and has been widely used in the rehabilitation of cerebrovascular diseases in Chinese hospitals. Many scholars have verified the effectiveness of acupuncture in treating aphasia by conducting clinical trials, meta-analyses, and other brain functional studies. Nevertheless, few studies have compared the efficacy of different intervention measures in treating aphasia and given the best treatment protocol. Therefore, choosing the best combination of acupuncture interventions has become a hot issue. Based on this, we searched the relevant literature and performed a network meta-analysis.

In this study, we compared the effects of different acupuncture interventions on improving language function in people with post-stroke aphasia. Our analysis showed that TA (SUCRA = 86%) was the best treatment for improving global efficiency (8.03; 95% CI, 3.62 to 17.81). At the same time, TA (SUCRA = 86%) also showed the highest probability of being the best treatment in oral expression according to the results of SUCRA. However, compared to SLT, TA has no significant statistical difference in improving expression ability (Table 4). Our result was in disagreement with the previous results by Tang et al. [126]; hence, further studies are still needed to identify the possible reasons for this discrepancy. Notably, although TA + BA ranked second with 83.2%, it showed statistically significant differences relative to SLT (18.08; 95% CI, 1.46 to 34.69). Therefore, these results indicated that BA + TA could also be served as an effective alternative treatment for PSA. BA + TA (SUCRA = 74.9%) showed the best outcome in increasing the comprehension score, but again, this was not statistically significant compared to SLT (Table 3). Even if BA (SUCRA = 67.1%, Rank 3) was superior to SLT with a significant difference (21.22; 95% CI, 0.12 to 42.32), we still recommend BA + TA as a treatment modality to improve speech comprehension in patients with PSA, because combined therapy was more consistent with the actual clinical situation, which might provide a potential therapeutic benefit for PSA patients. This was also confirmed by the review of Sun et al., which indicated that combination acupuncture therapies such as body acupuncture plus tongue acupuncture were more effective than body acupuncture in improving language ability [127]. BA + SA was the best treatment to improve aphasia patients’ repetition (25.55; 95% CI, 3.59 to 47.25) and denomination functions (29.47; 95% CI, 4.32 to 54.63). Although the curative effects of various measures discussed above are different under the evaluation of multiple indicators, TA, BA + TA, and BA + SA always ranked at the top of the SUCRA values. According to the above analysis, we inferred that TA, BA + TA, and BA + SA could be employed as complementary or alternative therapies to promote the functional recovery of PSA patients. Moreover, clinicians can make reasonable choices according to the characteristics of functional impairment in patients with PSA.

In the current literature, there are only a few reports on the mechanisms of tongue for treating neurological diseases such as visual impairment, cerebral palsy, and post-stroke dysphagia. Previous studies had shown that tongue acupuncture could improve the visual status of children with visual impairment and had a potential therapeutic effect on neural plasticity [128]. Another study showed that tongue acupuncture could enhance glucose metabolism in the frontal lobe, parietal lobe, temporal lobe, occipital cortex, and cerebellum of children with cerebral palsy in a short time and could be used as a candidate non-pharmacological intervention to improve functional recovery [129]. Although we were not able to find clinical studies on tongue treatment for PSA, it is apparent from the present studies that tongue acupuncture has substantial therapeutic potential as adjuvant therapy for post-stroke rehabilitation. An excellent recent review by Bono et al. describing the sensory motor the of tongue attracted our attention. They not only illustrated the importance of the extrinsic lingual muscles in language but also showed that different tongue regions could be somatotopically arranged in the cerebral cortex by electrocorticography studies, which had a high potential correlation with speech production and could enhance the control of fine movement required by language [130]. Thus, we speculate that the mechanism of tongue acupuncture improving aphasia symptoms may be related to the following reasons. First, tongue surfaces are innervated by numerous mechanoreceptors and nerves, which have high sensitivity for responding to multiple different stimuli. Second, tongue acupuncture may promote the regeneration of neurons and synapses in peri-infarcted regions, attenuate tissue damage, and finally improve overall neurological function. Lastly, long-term tongue acupuncture treatment may significantly improve the coordination and flexibility of the tongue muscle, reactivate primary language pathways, and enhance signal transduction between the tongue and the cerebral cortex to improve language abilities in PSA patients.

Body acupuncture often used with other alternative therapies is one of the most critical interventions in acupuncture therapies. Among the three treatment regimens we recommend, all the others included body acupuncture except for one using tongue acupuncture alone. The therapeutic benefits of acupuncture for PSA patients may be related to the activation of multiple language-related brain regions such as the frontal lobe, temporal lobe, and parietal lobe during treatment [131]. The study by Xie et al. showed that acupuncture could enhance the interregional interaction between the anterior cingulate cortex (ACC) and posterior cingulate cortex (PCC) to improve memory and cognitive ability [132]. In addition, stroke patients showed an increased level of mean flow velocity (MFV) [133], and the hemorheological parameters such as the index of erythrocyte aggregation and blood sedimentation were decreased after acupuncture treatment [134]. This may be an essential physiological mechanism by which acupuncture can treat cerebral ischemia and promote the recovery of neural function. There are few clinical trials and animal studies on treating aphasia with body acupuncture combined with tongue acupuncture. The combination of two interventions may play a synergistic role in improving language understanding and cognition, thus accelerating the rehabilitation of aphasia patients. The frequency, intensity, and duration of two interventions combined are further explored to verify this assumption.

Scalp acupuncture is a kind of acupuncture therapy that stimulates different brain functional areas to active neural issues and improves reflexivity for achieving the goals of treating diseases [135]. Some clinical studies have shown that scalp acupuncture has certain benefits in enhancing the language function and behavior of autistic children [136,137,138]. However, its related mechanism still needs to be further studied and explained. In recent years, RS-fMRI has been used widely in studying cerebral nerve activity and rehabilitation mechanisms of stroke patients treated with acupuncture. Liu et al. [139] performed RS-fMRI scanning on 30 patients with acute ischemic stroke who received scalp acupuncture treatment. The result showed that compared with the control group, the activity of neurons in the left angular gyrus, middle temporal gyrus, superior temporal gyrus, and fusiform gyrus were significantly increased, which were indispensable components of the language function network involved in language input and cognitive processing. Li et al., in their recent review, described in detail that scalp acupuncture and body acupuncture caused activity changes in the common brain regions during the treatment of aphasia patients, such as the right fusiform gyrus, the left superior frontal gyrus, and the left inferior temporal gyrus [140], which may be an important reason for the combination of the two acupuncture therapies to enhance the curative effect. Although few studies have directly revealed the mechanism of scalp acupuncture and body acupuncture in improving the repetition and naming of patients with post-stroke aphasia, these studies are of great significance for us to understand the mechanism of scalp acupuncture combined with body acupuncture in the treatment of post-stroke aphasia.

In summary, our study has made some efforts in rehabilitating patients with PSA. We found that tongue acupuncture, body acupuncture combined with tongue acupuncture, and body acupuncture combined with scalp acupuncture have a good therapeutic effect on aphasia after stroke, which can provide a reference for clinical practice. In addition, compared with expensive drugs and rehabilitation treatment, clinicians can try to take acupuncture as a good alternative or complementary therapy to promote aphasia rehabilitation after stroke.

### 4.2. Strengths and Limitations

First, our study included 69 studies and 5097 patients, which is a very large sample size. At the same time, we involved more than 10 kinds of treatment measures. We evaluated the impact of intervention measures from five aspects to provide more comprehensive evidence-based recommendations. Secondly, the research on acupuncture in treating aphasia mainly focuses on clinical trials and meta-analyses. In addition, there are a small number of fMRI imaging studies. Few studies have given recommendations for different acupuncture-related therapies in the treatment of PSA. We carried out the first network meta-analysis of acupuncture therapies for PSA and provided a preliminary experience for further detailed studies of this domain. However, this study also has some limitations, including the following. (I) Many studies did not specifically report the random method, assignment concealment, and reliability of the results. (II) Different treatment times, frequency, and protocols of acupuncture-related therapies included in the study may increase clinical heterogeneity. (III) Many studies only show the assessment of the four functional areas of aphasia, but not the evaluation of writing, reading, and other abilities. We could only analyze and provide general results based on these studies. In the subsequent study, we need to further introduce a variety of evaluation indicators to more comprehensively and objectively evaluate the effects of acupuncture therapies on PSA. (IV) Most of the participants included in this study were Chinese nationals, and the evaluation scale used was the ABC scale. Therefore, the research results were limited in universality and should be interpreted carefully. More well-designed studies with larger sample sizes and more ethnicities are required to minimize ethnic bias. (V) There is no minimum clinically important difference (MCID) available currently for ABC scores in patients with PSA. Future scholars should develop the MCIDs for such patients to better interpret the results of this paper in the clinical context.

### 4.3. Conclusions

Results of our network meta-analysis and SUCRA ranking showed that tongue acupuncture, body acupuncture + tongue acupuncture, and body acupuncture + scalp acupuncture seem to offer better advantages than other interventions for improving the language function in PSA patients. Moreover, it is noteworthy that our results are limited to the Chinese population, since all eligible studies are from China. Future well-designed studies with larger sample sizes and more ethnicities are required to further verify these findings.

## Figures and Tables

**Figure 1 brainsci-12-01282-f001:**
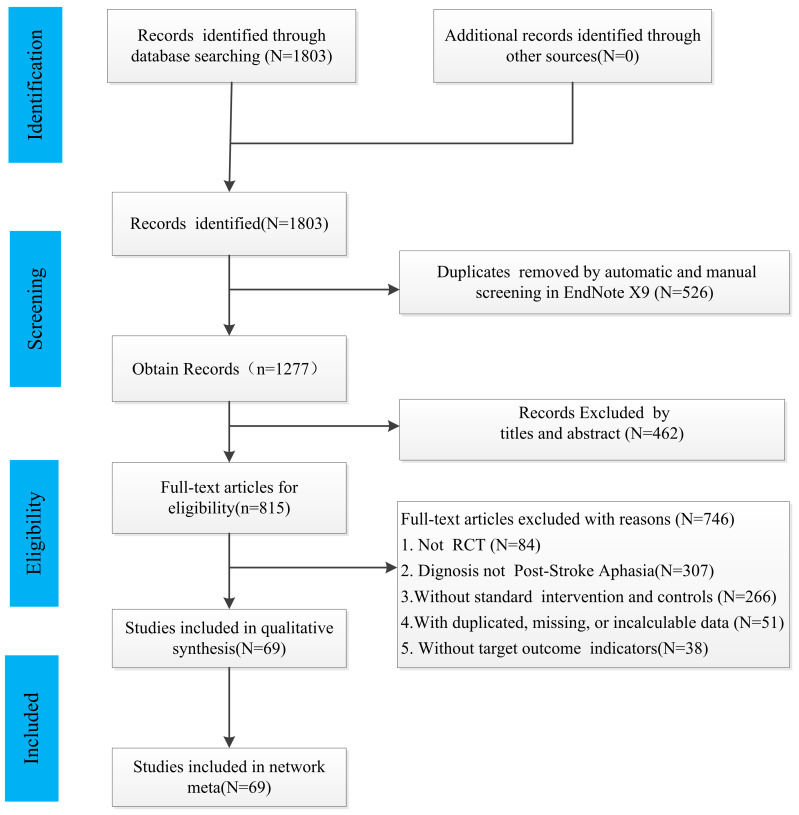
A PRISMA flow chart of literature research identification and selection process.

**Figure 7 brainsci-12-01282-f007:**
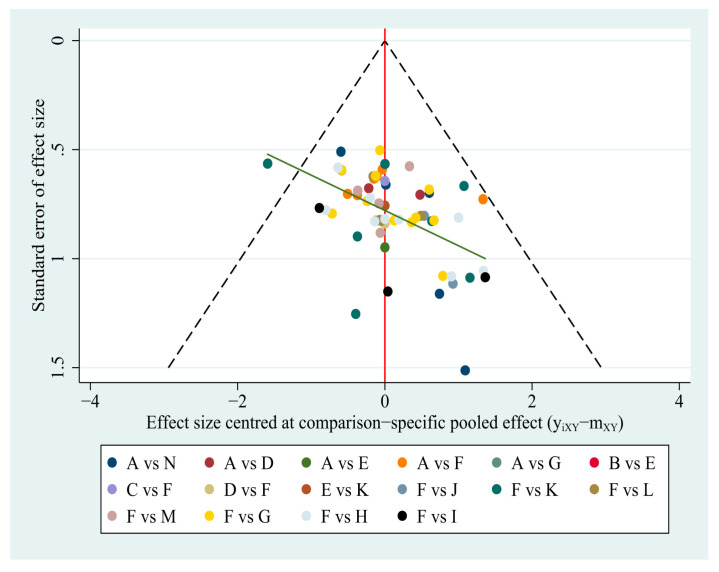
The “comparison-adjusted” funnel plot of total clinical effective rate.

**Table 1 brainsci-12-01282-t001:** Characteristics of the included studies.

Study	Sample Number		Gender (M/F)		Age		Therapy		Outcome	Reference
	T	C	T	C	T	C	T	C		
Bao W., 2020	40	40	25/15	13/17	61.30	62.48	SLT + BA + TA	SLT	1	[102]
Chen J., 2006	30	30	17/13	16/14	63.12	61.71	SLT + BA + SA + TA	SLT	1	[99]
Chen X., 2021	41	41	24/17	25/16	56.22	56.24	SLT + BA	SLT	1/2/4/5	[75]
Cheng X., 2018	20	20	14/6	13/7	63.15	63.34	SLT + SA	SLT	3/4/5	[61]
Dong X., 2020	56	56	30/26	33/23	60.85	62.57	SLT + SA	SLT	1	[62]
Dong Y., 2009	30	30	18/12	17/13	-	-	SLT + BA	SLT	1	[76]
Feng X., 2007	30	30	19/11	21/9	61	65	SLT + BA + SA	SLT	1	[90]
He J., 2021	56	56	26/30	31/25	55.67	56.12	SLT + BA	SLT	1	[77]
He A., 2014	30	30	14/16	13/17	55	54	SLT + TA	SLT	1	[105]
Hou B., 2018	50	50	26/24	24/26	33.6	34.2	SLT + BA + SA	SLT	1	[91]
Hou W., 2012	30	30	13/17	15/15	57.1	56.7	BA	SLT	1/2/3/4/5	[45]
Huang H., 2009	48	48	24/24	26/22	60.96	60.42	SLT + SA	SLT	1	[63]
Huang S., 2016	42	42	21/21	22/20	62	62	SLT + BA	BA	1	[78]
Jiang G., 2008	40	36	23/17	23/13	61.5	59.2	BA + TA	BA	1/2/3/4/5	[57]
Li L., 2019	30	30	19/11	17/13	63	64	TA	BA	3/4	[51]
Li L., 2020	30	30	17/13	16/14	56.07	54.47	BA + TA	BA	1/2/3/4/5	[58]
Li G., 2006	30	30	12/18	14/16	57.6	58.3	SLT + BA	SLT	1	[79]
Li Q., 2017	38	38	21/17	22/16	62.1	61.7	SLT + BA + SA	SLT	1	[92]
Li S., 2014	60	53	32/28	27/26	61.5	62.2	SLT + SA + TA	SLT	1	[70]
Li X., 2009	30	30	18/12	20/10	61.5	59.2	BA + TA	BA	1/2/3/4/5	[59]
Li Y., 2019	45	45	28/17	29/16	56.5	56.9	SLT + BA + SA	SLT	1	[93]
Li Z., 2019	40	40	23/17	25/15	57.7	58.03	SLT + BA + TA	SLT	1/2/3/4/5	[103]
Li Z.Z., 2019	41	41	21/20	24/17	66.49	67.05	SLT + TA	SLT	1	[106]
Li Z.P., 2004	32	30	20/12	18/12	60.5	61.8	TA	BA	1	[52]
Li Z.P., 2005	46	36	28/18	24/12	61.5	59.2	TA	BA	1	[53]
Liu J., 2018	50	50	27/23	31/19	57.98	58.05	BA	SLT	1	[46]
Liu J., 2018	36	36	20/16	18/18	57.68	58.01	BA	SLT	1	[47]
Liu J., 2018	36	36	19/17	18/18	56.87	58.01	BA	SLT	1/2/3/4/5	[48]
Liu L., 2006	30	28	22/8	18/10	-	-	SLT + BA	SLT	1	[80]
Liu X., 2011	32	32	19/13	21/11	65	65	TA	BA	1	[54]
Lu Q., 2010	32	32	22/10	26/6	50.5	45.3	SLT + BA + SA	SLT	1/2/3	[94]
Luo W., 2008	30	30	19/11	16/14	60.0	62.2	SLT + TA	SLT	2/3/4/5	[107]
Luo W., 2008	30	30	18/12	15/15	63.21	64.24	SLT + BA	SLT	2/3/4/5	[81]
Mi J., 2004	46	38	26/20	20/18	-	-	TA	BA	1	[55]
Qin L., 2018	40	40	-	-	-	-	SLT + BA	SLT	2/4/5	[82]
Qiu L., 2020	30	30	21/9	24/6	52.0	53.0	SLT + SA	SLT	1	[64]
Shi L., 2021	40	40	22/18	24/16	65.69	64.80	SLT + BA	SLT	1	[83]
Song C., 2017	31	30	20/11	18/12	61.0	60.1	SLT + TA	SLT	1	[108]
Song Z., 2019	35	35	16/19	15/20	58.35	57.25	SLT + SA	SLT	1/2/3/4/5	[65]
Tian H., 2012	58	40	30/28	20/20	-	-	SLT + BA	SLT	1	[84]
Tong X., 2017	42	42	24/18	22/20	61.8	61.0	SLT + SA + TA	SLT	1/2/4/5	[71]
Wang G., 2015	40	40	22/18	23/17	51.67	53.33	SLT + BA + SA	SLT	1/2/3/4/5	[95]
Wang L., 2011	40	40	20/20	22/18	63.4	64.5	SLT + BA + SA + TA	SLT	1/2/3/4/5	[100]
Wang M., 2016	40	40	24/16	25/15	63.6	63.5	SLT + SA + TA	SLT	1	[72]
Wang P., 1999	30	20	18/12	-	63	-	TA	BA	1	[56]
Wang Q., 2018	39	32	24/15	19/13	67.54	66.22	SLT + TA	SLT	2/3/4/5	[109]
Wang S., 2006	39	39	-	-	-	-	BA	SLT	1	[49]
Wang T., 2016	30	30	18/12	17/13	59.2	58.7	SLT + TA	SLT	1	[110]
Wang W., 2009	26	26	14/12	15/11	60.8	59.36	SLT + SA	SA	1/2/3	[66]
Wang Y., 2021	33	33	18/15	16/17	56.36	56.32	SLT + SA	SLT	1	[67]
Wei J., 2021	41	40	24/17	22/18	60.0	58.3	SLT + BA	SLT	1/2/3/4/5	[85]
Wei T., 2005	30	30	18/12	16/14	56.7	58.4	SLT + BA + SA + TA	SLT	1	[101]
Wen Z., 2019	45	45	21/24	23/22	60.41	60.36	SLT + SA + TA	SLT	1/2/3/4/5	[73]
Wu H., 2011	41	41	-	-	-	-	BA	SLT	1	[50]
Wu M., 2008	58	58	38/20	36/22	61	63	BA + TA	SLT	1	[60]
Xie M., 2015	22	22	12/10	14/8	52.1	50.4	SLT + BA + TA	SLT	1	[104]
Xu M., 2021	41	41	22/19	23/18	58.82	58.68	BA + SA	SA	1/2/3/4/5	[111]
Xu M., 2021	30	30	17/13	18/12	56.4	56.8	SLT + BA	SLT	1	[86]
Yang D., 2011	35	32	25/10	20/12	-	-	SA	BA	1/2/3/4/5	[113]
Yang X., 2015	35	35	-	-	-	-	SLT + BA + SA	SLT	1	[96]
Yu Q., 2018	30	30	18/12	17/13	57.06	58.52	SLT + BA	SLT	1	[87]
Zhang H., 2022	40	40	25/15	26/14	54.08	54.26	SLT + SA + TA	SLT	1	[74]
Zhang R., 2020	38	38	21/17	23/15	57.32	59.16	SLT + SA	SLT	1	[68]
Zhang S., 2010	24	23	15/9	13/10	62.7	63.2	SLT + BA + SA	SLT	1	[97]
Zhang Y., 2012	34	34	20/14	18/16	52	55	SLT + BA + SA	SLT	1	[98]
Zhang Y.J., 2020	40	40	-	-	-	-	SLT + SA	SLT	1	[69]
Zhao C., 2004	46	35	25/21	23/12	69	66	SLT + BA	SLT	1	[88]
Zhao D., 2021	48	48	26/22	27/21	61.79	62.42	SLT + BA	SLT	1	[89]
Zhou L., 2012	32	34	17/15	18/16	58.40	58.75	BA + SA + TA	SLT	1/2/3/4/5	[112]

Note: M, male; F, female; T, treatment group; C, control group; SA, scalp acupuncture; BA, body acupuncture; TA, tongue acupuncture; SLT, speech and language training; 1: the total effective rate; 2: the comprehension score; 3: the oral expression score; 4: the repetition score; 5: the denomination score.

**Table 2 brainsci-12-01282-t002:** Network analysis results of the total effective rate.

N	B	D	H	L	J	G	I	C	M	K	A	E	F
N	1.04 (0.12, 8.76)	0.74 (0.29, 1.93)	0.69 (0.26, 1.78)	0.62 (0.22, 1.76)	0.58 (0.18, 1.89)	0.55 (0.23, 1.31)	0.48 (0.13, 1.85)	0.48 (0.11, 2.12)	0.37 (0.13, 1.06)	0.37 (0.14, 0.95)	0.29 (0.15, 0.55)	0.22 (0.05, 0.88)	0.12 (0.06, 0.28)
**0.96 (0.11, 8.14)**	B	0.72 (0.08, 6.12)	0.66 (0.08, 5.35)	0.60 (0.07, 5.06)	0.56 (0.06, 5.10)	0.53 (0.07, 4.16)	0.47 (0.05, 4.64)	0.46 (0.04, 4.99)	0.36 (0.04, 3.03)	0.36 (0.05, 2.65)	0.28 (0.04, 2.14)	0.21 (0.04, 1.07)	0.12 (0.02, 0.91)
**1.34 (0.52, 3.49)**	**1.39 (0.16, 11.89)**	D	0.92 (0.35, 2.40)	0.83 (0.29, 2.38)	0.79 (0.24, 2.55)	0.74 (0.31, 1.78)	0.65 (0.17, 2.50)	0.64 (0.14, 2.85)	0.50 (0.17, 1.43)	0.50 (0.19, 1.29)	0.39 (0.19, 0.79)	0.29 (0.07, 1.20)	0.17 (0.08, 0.37)
**1.46 (0.56, 3.78)**	**1.51 (0.19, 12.25)**	**1.09 (0.42, 2.82)**	H	0.91 (0.39, 2.12)	0.85 (0.31, 2.33)	0.80 (0.42, 1.54)	0.71 (0.21, 2.34)	0.69 (0.18, 2.72)	0.54 (0.23, 1.28)	0.54 (0.26, 1.14)	0.42 (0.21, 0.86)	0.32 (0.08, 1.21)	0.18 (0.11, 0.31)
**1.61 (0.57, 4.56)**	**1.67 (0.20, 14.11)**	**1.20 (0.42, 3.41)**	**1.10 (0.47, 2.58)**	L	0.94 (0.32, 2.80)	0.88 (0.41, 1.92)	0.78 (0.22, 2.78)	0.77 (0.18, 3.20)	0.60 (0.23, 1.55)	0.59 (0.25, 1.40)	0.47 (0.21, 1.06)	0.35 (0.09, 1.42)	0.20 (0.10, 0.39)
**1.71 (0.53, 5.53)**	**1.77 (0.20, 16.03)**	**1.27 (0.39, 4.13)**	**1.17 (0.43, 3.21)**	**1.06 (0.36, 3.17)**	J	0.94 (0.36, 2.42)	0.83 (0.21, 3.30)	0.81 (0.18, 3.75)	0.63 (0.21, 1.90)	0.63 (0.23, 1.74)	0.50 (0.19, 1.33)	0.38 (0.08, 1.67)	0.21 (0.09, 0.50)
**1.82 (0.76, 4.36)**	**1.89 (0.24, 14.84)**	**1.36 (0.56, 3.27)**	**1.25 (0.65, 2.39)**	**1.13 (0.52, 2.46)**	**1.06 (0.41, 2.74)**	G	0.88 (0.28, 2.78)	0.87 (0.23, 3.25)	0.67 (0.31, 1.48)	0.67 (0.35, 1.31)	0.53 (0.29, 0.96)	0.40 (0.11, 1.43)	0.23 (0.15, 0.34)
**2.06 (0.54, 7.89)**	**2.14 (0.22, 21.24)**	**1.54 (0.40, 5.89)**	**1.41 (0.43, 4.68)**	**1.28 (0.36, 4.56)**	**1.21 (0.30, 4.80)**	**1.13 (0.36, 3.57)**	I	0.98 (0.19, 5.16)	0.76 (0.21, 2.74)	0.76 (0.23, 2.54)	0.60 (0.18, 1.94)	0.45 (0.09, 2.31)	0.26 (0.09, 0.76)
**2.10 (0.47, 9.35)**	**2.18 (0.20, 23.72)**	**1.56 (0.35, 6.98)**	**1.44 (0.37, 5.64)**	**1.31 (0.31, 5.45)**	**1.23 (0.27, 5.66)**	**1.15 (0.31, 4.33)**	**1.02 (0.19, 5.36)**	C	0.78 (0.19, 3.27)	0.78 (0.20, 3.06)	0.61 (0.16, 2.35)	0.46 (0.08, 2.67)	0.26 (0.07, 0.93)
**2.70 (0.94, 7.72)**	**2.80 (0.33, 23.76)**	**2.01 (0.70, 5.77)**	**1.85 (0.78, 4.37)**	**1.68 (0.64, 4.37)**	**1.58 (0.53, 4.74)**	**1.48 (0.67, 3.26)**	**1.31 (0.36, 4.69)**	**1.28 (0.31, 5.40)**	M	1.00 (0.42, 2.38)	0.78 (0.34, 1.80)	0.59 (0.15, 2.40)	0.34 (0.17, 0.67)
**2.71 (1.05, 7.00)**	**2.81 (0.38, 20.94)**	**2.02 (0.78, 5.24)**	**1.86 (0.88, 3.92)**	**1.68 (0.71, 3.97)**	**1.58 (0.57, 4.37)**	**1.49 (0.77, 2.89)**	**1.31 (0.39, 4.38)**	**1.29 (0.33, 5.08)**	**1.00 (0.42, 2.40)**	K	0.79 (0.39, 1.58)	0.59 (0.18, 1.96)	0.34 (0.20, 0.58)
**3.45 (1.82, 6.54)**	**3.58 (0.47, 27.37)**	**2.57 (1.26, 5.21)**	**2.36 (1.17, 4.78)**	**2.14 (0.94, 4.87)**	**2.02 (0.75, 5.39)**	**1.89 (1.05, 3.43)**	**1.67 (0.51, 5.43)**	**1.64 (0.43, 6.32)**	**1.28 (0.56, 2.94)**	**1.27 (0.63, 2.56)**	A	0.76 (0.22, 2.60)	0.43 (0.27, 0.69)
**4.56 (1.13, 18.33)**	**4.73 (0.94, 23.82)**	**3.39 (0.83, 13.85)**	**3.12 (0.83, 11.77)**	**2.83 (0.70, 11.40)**	**2.67 (0.60, 11.87)**	**2.50 (0.70, 8.98)**	**2.21 (0.43, 11.26)**	**2.17 (0.37, 12.55)**	**1.69 (0.42, 6.84)**	**1.68 (0.51, 5.53)**	**1.32 (0.38, 4.55)**	E	0.57 (0.17, 1.92)
**8.03 (3.62, 17.81)**	**8.32 (1.10, 63.13)**	**5.97 (2.68, 13.33)**	**5.50 (3.27, 9.25)**	**4.99 (2.55, 9.75)**	**4.69 (1.98, 11.10)**	**4.41 (2.98, 6.52)**	**3.89 (1.32, 11.44)**	**3.82 (1.08, 13.49)**	**2.97 (1.50, 5.89)**	**2.96 (1.73, 5.06)**	**2.33 (1.45, 3.74)**	**1.76 (0.52, 5.97)**	F

Efficacy of interventions are displayed in a league table. Treatments are ranked from best to worst along the leading diagonal. Data presented are the OR with 95% CI in the column-defining treatment compared to the row-defining treatment. For total clinical effective rate, ORs above 1 indicate higher efficacy in improving language function. Significant results are shown in bold. A: body acupuncture, B: body acupuncture + scalp acupuncture, C: body acupuncture + scalp acupuncture + tongue acupuncture, D: body acupuncture + tongue acupuncture, E: scalp acupuncture, F: speech and language training, G: speech and language training + body acupuncture, H: speech and language training + body acupuncture + scalp acupuncture, I: speech and language training + body acupuncture + scalp acupuncture + tongue acupuncture, J: speech and language training + body acupuncture + tongue acupuncture, K: speech and language training + scalp acupuncture, L: speech and language training + scalp acupuncture + tongue acupuncture, M: speech and language training + tongue acupuncture, and N: tongue acupuncture.

**Table 3 brainsci-12-01282-t003:** Network analysis results of the comprehensive sore.

D	B	A	N	G	E	L	J	M	H	K	C	I	F
D	0.06 (−48.01, 48.13)	−4.97 (−26.16, 16.22)	−3.78 (−40.58, 33.01)	−11.11 (−44.49, 22.28)	−10.95 (−48.64, 26.73)	−12.46 (−51.45, 26.52)	−16.23 (−58.38, 25.92)	−16.55 (−53.11, 20.01)	−18.76 (−55.36, 17.84)	−20.45 (−57.48, 16.59)	−24.60 (−66.62, 17.42)	−29.97 (−71.98, 12.04)	−26.19 (−56.09, 3.71)
−0.06 (−48.13, 48.01)	B	−5.03 (−48.19, 38.12)	−3.84 (−56.45, 48.76)	−11.17 (−56.97, 34.64)	−11.01 (−40.90, 18.88)	−12.52 (−62.54, 37.50)	−16.29 (−68.83, 36.25)	−16.61 (−64.77, 31.55)	−18.82 (−67.03, 29.38)	−20.51 (−60.21, 19.19)	−24.66 (−77.10, 27.78)	−30.03 (−82.46, 22.40)	−26.25 (−69.59, 17.08)
4.97 (−16.22, 26.16)	5.03 (−38.12, 48.19)	A	1.19 (−28.89, 31.27)	−6.13 (−31.93, 19.67)	−5.98 (−37.14, 25.19)	−7.49 (−40.21, 25.24)	−11.26 (−47.69, 25.18)	−11.57 (−41.37, 18.22)	−13.79 (−43.63, 16.06)	−15.47 (−45.85, 14.90)	−19.63 (−55.92, 16.66)	−25.00 (−61.28, 11.28)	−21.22 (−42.32, −0.12)
3.78 (−33.01, 40.58)	3.84 (−48.76, 56.45)	−1.19 (−31.27, 28.89)	N	−7.32 (−46.95, 32.31)	−7.17 (−50.48, 36.14)	−8.68 (−53.13, 35.77)	−12.45 (−59.69, 34.80)	−12.76 (−55.10, 29.58)	−14.98 (−57.35, 27.39)	−16.66 (−59.41, 26.09)	−20.82 (−67.95, 26.32)	−26.19 (−73.31, 20.94)	−22.41 (−59.15, 14.33)
11.11 (−22.28, 44.49)	11.17 (−34.64, 56.97)	6.13 (−19.67, 31.93)	7.32 (−32.31, 46.95)	G	0.15 (−34.58, 34.89)	−1.36 (−30.47, 27.75)	−5.12 (−38.36, 28.11)	−5.44 (−31.22, 20.34)	−7.66 (−33.50, 18.18)	−9.34 (−40.00, 21.31)	−13.49 (−46.57, 19.58)	−18.86 (−51.93, 14.20)	−15.09 (−29.96, −0.21)
10.95 (−26.73, 48.64)	11.01 (−18.88, 40.90)	5.98 (−25.19, 37.14)	7.17 (−36.14, 50.48)	−0.15 (−34.89, 34.58)	E	−1.51 (−41.64, 38.62)	−5.28 (−48.51, 37.96)	−5.59 (−43.38, 32.20)	−7.81 (−45.65, 30.03)	−9.49 (−35.64, 16.65)	−13.65 (−56.75, 29.46)	−19.02 (−62.12, 24.08)	−15.24 (−46.64, 16.17)
12.46 (−26.52, 51.45)	12.52 (−37.50, 62.54)	7.49 (−25.24, 40.21)	8.68 (−35.77, 53.13)	1.36 (−27.75, 30.47)	1.51 (−38.62, 41.64)	L	−3.77 (−42.62, 35.08)	−4.09 (−36.78, 28.61)	−6.30 (−39.05, 26.45)	−7.98 (−44.65, 28.68)	−12.14 (−50.85, 26.58)	−17.51 (−56.21, 21.20)	−13.73 (−38.75, 11.29)
16.23 (−25.92, 58.38)	16.29 (−36.25, 68.83)	11.26 (−25.18, 47.69)	12.45 (−34.80, 59.69)	5.12 (−28.11, 38.36)	5.28 (−37.96, 48.51)	3.77 (−35.08, 42.62)	J	−0.32 (−36.74, 36.10)	−2.53 (−39.00, 33.93)	−4.22 (−44.24, 35.81)	−8.37 (−50.28, 33.53)	−13.74 (−55.64, 28.16)	−9.96 (−39.68, 19.76)
16.55 (−20.01, 53.11)	16.61 (−31.55, 64.77)	11.57 (−18.22, 41.37)	12.76 (−29.58, 55.10)	5.44 (−20.34, 31.22)	5.59 (−32.20, 43.38)	4.09 (−28.61, 36.78)	0.32 (−36.10, 36.74)	M	−2.21 (−32.04, 27.61)	−3.90 (−37.98, 30.18)	−8.05 (−44.33, 28.22)	−13.42 (−49.69, 22.84)	−9.65 (−30.70, 11.41)
18.76 (−17.84, 55.36)	18.82 (−29.38, 67.03)	13.79 (−16.06, 43.63)	14.98 (−27.39, 57.35)	7.66 (−18.18, 33.50)	7.81 (−30.03, 45.65)	6.30 (−26.45, 39.05)	2.53 (−33.93, 39.00)	2.21 (−27.61, 32.04)	H	−1.68 (−35.82, 32.45)	−5.84 (−42.16, 30.48)	−11.21 (−47.52, 25.10)	−7.43 (−28.56, 13.70)
20.45 (−16.59, 57.48)	20.51 (−19.19, 60.21)	15.47 (−14.90, 45.85)	16.66 (−26.09, 59.41)	9.34 (−21.31, 40.00)	9.49 (−16.65, 35.64)	7.98 (−28.68, 44.65)	4.22 (−35.81, 44.24)	3.90 (−30.18, 37.98)	1.68 (−32.45, 35.82)	K	−4.15 (−44.04, 35.74)	−9.52 (−49.40, 30.36)	−5.75 (−32.56, 21.07)
24.60 (−17.42, 66.62)	24.66 (−27.78, 77.10)	19.63 (−16.66, 55.92)	20.82 (−26.32, 67.95)	13.49 (−19.58, 46.57)	13.65 (−29.46, 56.75)	12.14 (−26.58, 50.85)	8.37 (−33.53, 50.28)	8.05 (−28.22, 44.33)	5.84 (−30.48, 42.16)	4.15 (−35.74, 44.04)	C	−5.37 (−47.14, 36.40)	−1.59 (−31.13, 27.95)
29.97 (−12.04, 71.98)	30.03 (−22.40, 82.46)	25.00 (−11.28, 61.28)	26.19 (−20.94, 73.31)	18.86 (−14.20, 51.93)	19.02 (−24.08, 62.12)	17.51 (−21.20, 56.21)	13.74 (−28.16, 55.64)	13.42 (−22.84, 49.69)	11.21 (−25.10, 47.52)	9.52 (−30.36, 49.40)	5.37 (−36.40, 47.14)	I	3.78 (−25.75, 33.31)
26.19 (−3.71, 56.09)	26.25 (−17.08, 69.59)	**21.22** **(0.12, 42.32)**	22.41(−14.33, 59.15)	**15.09** **(0.21, 29.96)**	15.24 (−16.17, 46.64)	13.73 (−11.29, 38.75)	9.96 (−19.76, 39.68)	9.65 (−11.41, 30.70)	7.43 (−13.70, 28.56)	5.75 (−21.07, 32.56)	1.59 (−27.95, 31.13)	−3.78 (−33.31, 25.75)	F

Efficacy of interventions are displayed in a league table. Treatments are ranked from best to worst along the leading diagonal. Data presented are the MD with 95% CI in the column-defining treatment compared to the row-defining treatment. For ABC sores, MD above 0 favors the treatment in column in improving ABC scores. Significant results are shown in bold. A: body acupuncture, B: body acupuncture + scalp acupuncture, C: body acupuncture + scalp acupuncture + tongue acupuncture, D: body acupuncture + tongue acupuncture, E: scalp acupuncture, F: speech and language training, G: speech and language training + body acupuncture, H: speech and language training + body acupuncture + scalp acupuncture, I: speech and language training + body acupuncture + scalp acupuncture + tongue acupuncture, J: speech and language training + body acupuncture + tongue acupuncture, K: speech and language training + scalp acupuncture, L: speech and language training + scalp acupuncture + tongue acupuncture, M: speech and language training + tongue acupuncture, N: tongue acupuncture.

**Table 4 brainsci-12-01282-t004:** Network analysis results of the oral expression sore.

N	D	B	A	K	G	M	L	J	E	I	C	H	F
N	−3.64 (−26.07, 18.80)	−7.79 (−39.85, 24.26)	−12.70 (−32.23, 6.83)	−13.73 (−40.61, 13.15)	−14.57 (−41.47, 12.33)	−15.63 (−42.53, 11.27)	−16.61 (−46.74, 13.53)	−16.98 (−42.12, 8.15)	−17.30 (−42.82, 8.22)	−20.71 (−50.69, 9.27)	−21.05 (−50.98, 8.88)	−21.31 (−51.26, 8.64)	−21.71 (−44.86, 1.43)
3.64 (−18.80, 26.07)	D	−4.16 (−31.86, 23.55)	−9.06 (−20.10, 1.97)	−10.09 (−31.61, 11.42)	−10.93 (−32.47, 10.61)	−11.99 (−33.53, 9.55)	−12.97 (−38.43, 12.49)	−13.35 (−32.63, 5.94)	−13.66 (−33.45, 6.12)	−17.07 (−42.35, 8.21)	−17.41 (−42.63, 7.81)	−17.67 (−42.92, 7.57)	−18.08 (−34.69, −1.46)
7.79 (−24.26, 39.85)	4.16 (−23.55, 31.86)	B	−4.91 (−30.32, 20.51)	−5.94 (−35.13, 23.26)	−6.78 (−35.98, 22.43)	−7.84 (−37.04, 21.37)	−8.82 (−41.03, 23.39)	−9.19 (−34.22, 15.84)	−9.51 (−28.91, 9.90)	−12.92 (−44.99, 19.15)	−13.26 (−45.28, 18.76)	−13.52 (−45.56, 18.52)	−13.92 (−39.71, 11.87)
12.70 (−6.83, 32.23)	9.06 (−1.97, 20.10)	4.91 (−20.51, 30.32)	A	−1.03 (−19.50, 17.44)	−1.87 (−20.36, 16.63)	−2.93 (−21.42, 15.57)	−3.91 (−26.85, 19.04)	−4.28 (−20.10, 11.54)	−4.60 (−21.02, 11.82)	−8.01 (−30.75, 14.74)	−8.35 (−31.02, 14.33)	−8.61 (−31.31, 14.10)	−9.01 (−21.43, 3.41)
13.73 (−13.15, 40.61)	10.09 (−11.42, 31.61)	5.94 (−23.26, 35.13)	1.03 (−17.44, 19.50)	K	−0.84 (−20.20, 18.53)	−1.90 (−21.26, 17.47)	−2.88 (−26.53, 20.78)	−3.25 (−21.73, 15.22)	−3.57 (−25.39, 18.25)	−6.98 (−30.44, 16.48)	−7.32 (−30.71, 16.07)	−7.58 (−31.00, 15.84)	−7.98 (−21.66, 5.70)
14.57 (−12.33, 41.47)	10.93 (−10.61, 32.47)	6.78 (−22.43, 35.98)	1.87 (−16.63, 20.36)	0.84 (−18.53, 20.20)	G	−1.06 (−20.45, 18.32)	−2.04 (−25.71, 21.63)	−2.42 (−20.91, 16.08)	−2.73 (−24.56, 19.10)	−6.14 (−29.62, 17.34)	−6.48 (−29.89, 16.93)	−6.74 (−30.18, 16.70)	−7.15 (−20.86, 6.56)
15.63 (−11.27, 42.53)	11.99 (−9.55, 33.53)	7.84 (−21.37, 37.04)	2.93 (−15.57, 21.42)	1.90 (−17.47, 21.26)	1.06 (−18.32, 20.45)	M	−0.98 (−24.65, 22.69)	−1.35 (−19.85, 17.14)	−1.67 (−23.50, 20.16)	−5.08 (−28.56, 18.40)	−5.42 (−28.83, 17.99)	−5.68 (−29.12, 17.76)	−6.09 (−19.79, 7.62)
16.61 (−13.53, 46.74)	12.97 (−12.49, 38.43)	8.82 (−23.39, 41.03)	3.91 (−19.04, 26.85)	2.88 (−20.78, 26.53)	2.04 (−21.63, 25.71)	0.98 (−22.69, 24.65)	L	−0.37 (−23.32, 22.57)	−0.69 (−26.41, 25.02)	−4.10 (−31.22, 23.02)	−4.44 (−31.50, 22.62)	−4.70 (−31.79, 22.39)	−5.10 (−24.40, 14.19)
16.98 (−8.15, 42.12)	13.35 (−5.94, 32.63)	9.19 (−15.84, 34.22)	4.28 (−11.54, 20.10)	3.25 (−15.22, 21.73)	2.42 (−16.08, 20.91)	1.35 (−17.14, 19.85)	0.37 (−22.57, 23.32)	J	−0.32 (−16.13, 15.49)	−3.73 (−26.47, 19.02)	−4.07 (−26.74, 18.61)	−4.33 (−27.03, 18.38)	−4.73 (−17.15, 7.69)
17.30 (−8.22, 42.82)	13.66 (−6.12, 33.45)	9.51 (−9.90, 28.91)	4.60 (−11.82, 21.02)	3.57 (−18.25, 25.39)	2.73 (−19.10, 24.56)	1.67 (−20.16, 23.50)	0.69 (−25.02, 26.41)	0.32 (−15.49, 16.13)	E	−3.41 (−28.95, 22.13)	−3.75 (−29.22, 21.72)	−4.01 (−29.51, 21.49)	−4.41 (−21.41, 12.58)
20.71 (−9.27, 50.69)	17.07 (−8.21, 42.35)	12.92 (−19.15, 44.99)	8.01 (−14.74, 30.75)	6.98 (−16.48, 30.44)	6.14 (−17.34, 29.62)	5.08(−18.40, 28.56)	4.10 (−23.02, 31.22)	3.73 (−19.02, 26.47)	3.41 (−22.13, 28.95)	I	−0.34 (−27.23, 26.55)	−0.60 (−27.52, 26.32)	−1.00 (−20.06, 18.06)
21.05 (−8.88, 50.98)	17.41 (−7.81, 42.63)	13.26 (−18.76, 45.28)	8.35 (−14.33, 31.02)	7.32 (−16.07, 30.71)	6.48 (−16.93, 29.89)	5.42 (−17.99, 28.83)	4.44 (−22.62, 31.50)	4.07 (−18.61, 26.74)	3.75 (−21.72, 29.22)	0.34 (−26.55, 27.23)	C	−0.26 (−27.12, 26.60)	−0.66 (−19.64, 18.31)
21.31 (−8.64, 51.26)	17.67 (−7.57, 42.92)	13.52 (−18.52, 45.56)	8.61 (−14.10, 31.31)	7.58 (−15.84, 31.00)	6.74 (−16.70, 30.18)	5.68 (−17.76, 29.12)	4.70 (−22.39, 31.39)	4.33 (−18.38, 27.03)	4.01 (−21.49, 29.51)	0.60 (−26.32, 27.52)	0.26 (−26.60, 27.12)	H	−0.40 (−19.41, 18.61)
21.71 (−1.43, 44.86)	**18.08** **(1.46, 34.69)**	13.92 (−11.87, 39.71)	9.01 (−3.41, 21.43)	7.98 (−5.70, 21.66)	7.15 (−6.56, 20.86)	6.09 (−7.62, 19.79)	5.10 (−14.19, 24.40)	4.73 (−7.69, 17.15)	4.41 (−12.58, 121.41)	1.00 (−18.06, 20.06)	0.66 (−18.31, 19.64)	0.40 (−18.61, 19.41)	F

Efficacy of interventions are displayed in a league table. Treatments are ranked from best to worst along the leading diagonal. Data presented are the MD with 95% CI in the column-defining treatment compared to the row-defining treatment. For ABC sores, MD above 0 favors the treatment in column in improving ABC scores. Significant results are shown in bold. A: body acupuncture, B: body acupuncture + scalp acupuncture, C: body acupuncture + scalp acupuncture + tongue acupuncture, D: body acupuncture + tongue acupuncture, E: scalp acupuncture, F: speech and language training, G: speech and language training + body acupuncture, H: speech and language training + body acupuncture + scalp acupuncture + tongue acupuncture, I: speech and language training + body acupuncture + tongue acupuncture, J: speech and language training + scalp acupuncture, K: speech and language training + scalp acupuncture + body acupuncture, L: speech and language training + scalp acupuncture + tongue acupuncture, M: speech and language training + tongue acupuncture, N: tongue acupuncture.

**Table 5 brainsci-12-01282-t005:** Network analysis results of the repetition score.

B	M	D	E	A	K	G	I	L	J	C	H	F
B	−4.06 (−27.23, 19.12)	−5.57 (−26.61, 15.46)	−11.11 (−22.75, 0.54)	−13.00 (−33.02, 7.02)	−13.97 (−37.32, 9.37)	−14.72 (−37.40, 7.96)	−17.40 (−40.71, 5.90)	−19.06 (−42.40, 4.28)	−23.53 (−47.22, 0.17)	−24.50 (−48.93, −0.07)	−24.60 (−49.21, 0.01)	−25.55 (−47.52, −3.59)
4.06 (−19.12, 27.23)	M	−1.52 (−14.87, 11.83)	−7.05 (−27.09, 12.99)	−8.94 (−20.62, 2.74)	−9.92 (−26.67, 6.84)	−10.66 (−26.48, 5.15)	−13.35 (−30.04, 3.35)	−15.00 (−31.74, 1.74)	−19.47 (−36.70, −2.23)	−20.45 (−38.68, −2.21)	−20.55 (−39.02, −2.07)	−21.50 (−36.26, −6.74)
5.57 (−15.46, 26.61)	1.52 (−11.83, 14.87)	D	−5.53 (−23.06, 11.99)	−7.42 (−13.89, −0.96)	−8.40 (−22.04, 5.24)	−9.15 (−21.61, 3.32)	−11.83 (−25.39, 1.73)	−13.49 (−27.10, 0.13)	−17.95 (−32.17, −3.73)	−18.93 (−34.35, −3.51)	−19.03 (−34.73, −3.33)	−19.98 (−31.08, −8.88)
11.11 (−0.54, 22.75)	7.05 (−12.99, 27.09)	5.53 (−11.99, 23.06)	E	−1.89 (−18.18, 14.39)	−2.87 (−23.11, 17.37)	−3.61 (−23.08, 15.85)	−6.30 (−26.48, 13.89)	−7.95 (−28.18, 12.27)	−12.42 (−33.05, 8.22)	−13.40 (−34.87, 8.08)	−13.50 (−35.18, 8.18)	−14.45 (−33.07, 4.17)
13.00 (−7.02, 33.02)	8.94 (−2.74, 20.62)	**7.42** **(0.96, 13.89)**	1.89 (−14.39, 18.18)	A	−0.98 (−12.99, 11.04)	−1.72 (−12.39, 8.94)	−4.41 (−16.33, 7.52)	−6.06 (−18.05, 5.93)	−10.53 (−23.20, 2.15)	−11.50 (−25.51, 2.50)	−11.60 (−25.92, 2.71)	−12.56 (−21.58, −3.53)
13.97 (−9.37, 37.32)	9.92 (−6.84, 26.67)	8.40 (−5.24, 22.04)	2.87 (−17.37, 23.11)	0.98 (−11.04, 12.99)	K	−0.75 (−10.53, 9.04)	−3.43 (−14.59, 7.73)	−5.09 (−16.28, 6.11)	−9.55 (−21.46, 2.36)	−10.53 (−23.89, 2.83)	−10.63 (−24.31, 3.05)	−11.58 (−19.57, −3.59)
14.72 (−7.96, 37.40)	10.66 (−5.15, 26.48)	9.15 (−3.32, 21.61)	3.61 (−15.85, 23.08)	1.72 (−8.94, 12.39)	0.75 (−9.04, 10.53)	G	−2.68 (−12.31, 6.94)	−4.34 (−13.98, 5.30)	−8.80 (−19.27, 1.66)	−9.78 (−21.88, 2.32)	−9.88 (−22.34, 2.57)	−10.83 (−16.47, −5.19)
17.40 (−5.90, 40.71)	13.35 (−3.35, 30.04)	11.83 (−1.73, 25.39)	6.30 (−13.89, 26.48)	4.41 (−7.52, 16.33)	3.43 (−7.73, 14.59)	2.68 (−6.94, 12.31)	I	−1.66 (−12.71, 9.39)	−6.12 (−17.89, 5.65)	−7.10 (−20.34, 6.15)	−7.20 (−20.77, 6.37)	−8.15 (−15.95, −0.35)
19.06 (−4.28, 42.40)	15.00 (−1.74, 31.74)	13.49 (−0.13, 27.10)	7.95 (−12.27, 28.18)	6.06 (−5.93, 18.05)	5.09 (−6.11, 16.28)	4.34 (−5.30, 13.98)	1.66 (−9.39, 12.71)	L	−4.46 (−16.24, 7.31)	−5.44 (−18.71, 7.82)	−5.54 (−19.13, 8.05)	−6.49 (−14.32, 1.33)
23.53 (−0.17, 47.22)	**19.47** **(2.23, 36.70)**	**17.95** **(3.73, 32.17)**	12.42 (−8.22, 33.05)	10.53 (−2.15, 23.20)	9.55 (−2.36, 21.46)	8.80 (−1.66, 19.27)	6.12 (−5.65, 17.89)	4.46 (−7.31, 16.24)	J	−0.98 (−14.85, 12.89)	−1.08 (−15.26, 13.11)	−2.03 (−10.85, 6.79)
**24.50** **(0.07, 48.93)**	**20.45** **(2.21, 38.68)**	**18.93** **(3.51, 34.35)**	13.40 (−8.08, 34.87)	11.50 (−2.50, 25.51)	10.53 (−2.83, 23.89)	9.78 (−2.32, 21.88)	7.10 (−6.15, 20.34)	5.44 (−7.82, 18.71)	0.98 (−12.89, 14.85)	C	−0.10 (−15.53, 15.33)	−1.05 (−11.76, 9.65)
24.60 (−0.01, 49.21)	**20.55** **(2.07, 39.02)**	**19.03** **(3.33, 34.73)**	13.50 (−8.18, 35.18)	11.60 (−2.71, 25.92)	10.63 (−3.05, 24.31)	9.88 (−2.57, 22.34)	7.20 (−6.37, 20.77)	5.54 (−8.05, 19.13)	1.08 (−13.11, 15.26)	0.10 (−15.33, 15.53)	H	−0.95 (−12.06, 10.15)
**25.55** **(3.59, 47.52)**	**21.50** **(6.74, 36.26)**	**19.98** **(8.88, 31.08)**	14.45 (−4.17, 33.07)	**12.56** **(3.53, 21.58)**	**11.58** **(3.59, 19.57)**	**10.83** **(5.19, 16.47)**	**8.15** **(0.35, 15.95)**	6.49 (−1.33, 14.32)	2.03 (−6.79, 10.85)	1.05 (−9.65, 11.76)	0.95 (−10.15, 12.06)	F

Efficacy of interventions are displayed in a league table. Treatments are ranked from best to worst along the leading diagonal. Data presented are the MD with 95% CI in the column-defining treatment compared to the row-defining treatment. For ABC sores, MD above 0 favors the treatment in column in improving ABC scores. Significant results are shown in bold. A: body acupuncture, B: body acupuncture + scalp acupuncture, C: body acupuncture + scalp acupuncture + tongue acupuncture, D: body acupuncture + tongue acupuncture, E: scalp acupuncture, F: speech and language training, G: speech and language training + body acupuncture, H: speech and language training + body acupuncture + scalp acupuncture, I: speech and language training + body acupuncture + scalp acupuncture + tongue acupuncture, J: speech and language training + body acupuncture + tongue acupuncture, K: speech and language training + scalp acupuncture, L: speech and language training + scalp acupuncture + tongue acupuncture, M: speech and language training + tongue acupuncture.

**Table 6 brainsci-12-01282-t006:** Network analysis results of the denomination score.

B	D	A	E	G	L	H	M	J	C	I	F	K
B	−6.81 (−28.94, 15.31)	−11.96 (−33.01, 9.09)	−10.07 (−22.66, 2.53)	−18.83 (−42.95, 5.30)	−19.63 (−44.45, 5.18)	−21.02 (−47.14, 5.09)	−22.30 (−47.27, 2.67)	−23.60 (−49.81, 2.60)	−28.08 (−54.15, −2.02)	−28.44 (−54.51, −2.38)	−28.65 (−52.01, −5.29)	−29.47 (−54.63, −4.32)
6.81 (−15.31, 28.94)	D	−5.15 (−11.95, 1.66)	−3.25 (−21.44, 14.94)	−12.01 (−25.61, 1.59)	−12.82 (−27.61, 1.96)	−14.21 (−31.08, 2.66)	−15.49 (−30.53, −0.45)	−16.79 (−33.81, 0.23)	−21.27 (−38.08, −4.47)	−21.63 (−38.43, −4.83)	−21.83 (−34.02, −9.65)	−22.66 (−38.02, −7.31)
11.96 (−9.09, 33.01)	5.15 (−1.66, 11.95)	A	1.89 (−14.98, 18.76)	−6.87 (−18.66, 4.92)	−7.68 (−20.81, 5.46)	−9.07 (−24.51, 6.38)	−10.34 (−23.77, 3.08)	−11.65 (−27.25, 3.96)	−16.13 (−31.50, −0.75)	−16.49 (−31.86, −1.11)	−16.69 (−26.81, −6.56)	−17.52 (−31.29, −3.74)
10.07 (−2.53, 22.66)	3.25 (−14.94, 21.44)	−1.89 (−18.76, 14.98)	E	−8.76 (−29.34, 11.82)	−9.57 (−30.95, 11.81)	−10.96 (−33.83, 11.92)	−12.23 (−33.79, 9.33)	−13.54 (−36.52, 9.45)	−18.02 (−40.84, 4.81)	−18.38 (−41.20, 4.45)	−18.58 (−38.25, 1.10)	−19.41 (−41.19, 2.37)
18.83 (−5.30, 42.95)	12.01 (−1.59, 25.61)	6.87 (−4.92, 18.66)	8.76 (−11.82, 29.34)	G	−0.81 (−11.12, 9.50)	−2.20 (−15.31, 10.92)	−3.47 (−14.01, 7.06)	−4.78 (−18.08, 8.53)	−9.26 (−22.29, 3.77)	−9.62 (−22.64, 3.41)	−9.82 (−15.81, −3.83)	−10.65 (−21.71, 0.42)
19.63 (−5.18, 44.45)	12.82 (−1.96, 27.61)	7.68 (−5.46, 20.81)	9.57 (−11.81, 30.95)	0.81 (−9.50, 11.12)	L	−1.39 (−15.76, 12.98)	−2.67 (−14.74, 9.41)	−3.97 (−18.52, 10.58)	−8.45 (−22.74, 5.84)	−8.81 (−23.10, 5.48)	−9.01 (−17.40, −0.62)	−9.84 (−22.37, 2.69)
21.02 (−5.09, 47.14)	14.21 (−2.66, 31.08)	9.07 (−6.38, 24.51)	10.96 (−11.92, 33.83)	2.20 (−10.92, 15.31)	1.39 (−12.98, 15.76)	H	−1.28 (−15.82, 13.27)	−2.58 (−19.23, 14.07)	−7.06 (−23.49, 9.37)	−7.42 (−23.85, 9.01)	−7.62 (−19.29, 4.05)	−8.45 (−23.37, 6.48)
22.30 (−2.67, 47.27)	**15.49** **(0.45, 30.53)**	10.34 (−3.08, 23.77)	12.23 (−9.33, 33.79)	3.47 (−7.06, 14.01)	2.67 (−9.41, 14.74)	1.28 (−13.27, 15.82)	M	−1.30 (−16.02, 13.41)	−5.78 (−20.25, 8.68)	−6.14 (−20.60, 8.32)	−6.35 (−15.03, 2.33)	−7.17 (−19.89, 5.54)
23.60 (−2.60, 49.81)	16.79 (−0.23, 33.81)	11.65 (−3.96, 27.25)	13.54 (−9.45, 36.52)	4.78 (−8.53, 18.08)	3.97 (−10.58, 18.52)	2.58 (−14.07, 19.23)	1.30 (−13.41, 16.02)	J	−4.48 (−21.06, 12.10)	−4.84 (−21.42, 11.74)	−5.04 (−16.92, 6.84)	−5.87 (−20.96, 9.22)
**28.08** **(2.02, 54.15)**	**21.27** **(4.47, 38.08)**	**16.13** **(0.75, 31.50)**	18.02 (−4.81, 40.84)	9.26 (−3.77, 22.29)	8.45 (−5.84, 22.74)	7.06 (−9.37, 23.49)	5.78 (−8.68, 20.25)	4.48 (−12.10, 21.06)	C	−0.36 (−16.72, 16.00)	−0.56 (−12.13, 11.01)	−1.39 (−16.24, 13.46)
**28.44** **(2.38, 54.51)**	**21.63** **(4.83, 38.43)**	**16.49** **(1.11, 31.86)**	18.38 (−4.45, 41.20)	9.62 (−3.41, 22.64)	8.81 (−5.48, 23.10)	7.42 (−9.01, 23.85)	6.14 (−8.32, 20.60)	4.84 (−11.74, 21.42)	0.36 (−16.00, 16.72)	I	−0.20 (−11.77, 11.36)	−1.03 (−15.87, 13.82)
**28.65** **(5.29, 52.01)**	**21.83** **(9.65, 34.02)**	**16.69** **(6.56, 26.81)**	18.58 (−1.10, 38.25)	**9.82** **(3.83, 15.81)**	**9.01** **(0.62, 17.40)**	7.62 (−4.05, 19.29)	6.35 (−2.33, 15.03)	5.04 (−6.84, 16.92)	0.56 (−11.01, 12.13)	0.20 (−11.36, 11.77)	F	−0.83 (−10.13, 8.48)
**29.47** **(4.32, 54.63)**	**22.66** **(7.31, 38.02)**	**17.52** **(3.74, 31.29)**	19.41 (−2.37, 41.19)	10.65 (−0.42, 21.71)	9.84 (−2.69, 22.37)	8.45 (−6.48, 23.37)	7.17 (−5.54, 19.89)	5.87 (−9.22, 20.96)	1.39 (−13.46, 16.24)	1.03 (−13.82, 15.87)	0.83 (−8.48, 10.13)	K

Efficacy of interventions are displayed in a league table. Treatments are ranked from best to worst along the leading diagonal. Data presented are the MD with 95% CI in the column-defining treatment compared to the row-defining treatment. For ABC sores, MD above 0 favored the treatment in column in improving ABC scores. Significant results are shown in bold. A: body acupuncture, B: body acupuncture + scalp acupuncture, C: body acupuncture + scalp acupuncture + tongue acupuncture, D: body acupuncture + tongue acupuncture, E: scalp acupuncture, F: speech and language training, G: speech and language training + body acupuncture, H: speech and language training + body acupuncture + scalp acupuncture, I: speech and language training + body acupuncture + scalp acupuncture + tongue acupuncture, J: speech and language training + body acupuncture + tongue acupuncture, K: speech and language training + scalp acupuncture, L: speech and language training + scalp acupuncture + tongue acupuncture, M: speech and language training + tongue acupuncture.

## Data Availability

The data set and Stata codes that support the results of this study are shared as publicly accessible Appendix A (https://osf.io/vej78/ (accessed on 16 September 2022)) for further exploration.

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
