# Peer review of "Effects of Acupuncture-Related Therapies in the Rehabilitation of Patients with Post-Stroke Aphasia—A Network Meta-Analysis of Randomized Controlled Trials"

_brainsci, 2022, doi:10.3390/brainsci12101282_

Round 1

Reviewer 1 Report

Dear Authors; I found this work an interesting endeavour to apply NMA statistical framework to analysis effects of acupuncture related therapies in the patients' rehabiliation. It needs some minor work to arrive to publication. Regards. P.S.

[1] Writing:

1-1. Add the list of use abbreviations in the text right before Reference Section.

Example:

Abbreviations:

BA: Body Acupuncture; etc.

1-2. Add comments for Section 3.4.1. 

1-3. Figures 2-6 (A):  Is the size of the blue circles demonstrating the size of the trial? If so, need to add legend to the figures showing this size of trial level. 

1-4. Resectionize sections "5.Limitations & Strengths" and "6.Conclusion" as subsections of the "4.Discussion". The current status has organizational problem. Something like this:

4.Discussion 4.1. This work <- put material in current section 4 in it. 4.2. Strengths & Limitations 4.3. Future Work  <-- add a paragraph for this 4.4. Conclusions

[2] Statistical:

2-1 Random Effects NMA: Defend your choice of more complex random effects NMA over fixed effects NMA. Add a paragraph to the text.

2-2 Frequentist RE NMA: Defend your choice of Frequentist RE NMA over Bayesian RE NMA. Add a paragraph to the text.

2-3 Missing Formulae:  Report your Random Effects NMA formula in section "2.7. Statistical Analysis". 

Reviewer 2 Report

1.     The search strategy and keywords used in the Chinese databases should be provided as supplementary materials.

2.     The standard version of the ABC assessment should be cited.

3.     The definition of effective rate should be clearly drafted.

4.     The treatment details of the treatments should be presented.

5.     The baseline scores of the comprehension, verbal expression, repetition, and denomination functions of each included study should be presented.

6.     Why do the “reading” and “writing” scores in the ABC assessment are not included in the present study?

7.     “Figure X” in lines 356 and 362 should be revised.

8.     The results demonstrated the differences between each treatment combination. However, SMD is difficult to interpret the clinical significance. Please transfer the results into clinical understanding. Are the best treatment effects in each NMA significantly better than the minimal clinically important difference (MCID) of ABC assessment?

9.     Since the authors chose the ABC assessment as an outcome, the results and conclusions are restricted to Chinese user patients. This point should be presented in the manuscript.

Reviewer 3 Report

Dear authors, the manuscript is intriguing and niche, well structured however with methodological and expository concerns to address

Nonetheless it is a systematic review and network meta-analysis

Remove the software from the abstract, if possible, add a systematic review registration codeDescribe abbreviation SUCRA

24 I recommend softening the results as it is always an advanced type of statistic with inherent biases to pairwised. The results seem to report, in conclusion it appears..

28 Removing any recommendations is not a guideline. Report the results, please

49 Virtual reality .. describe multiple rehabilitation approaches

67 references missing

98 there are PRISMA-NMA checklists

I would define the network plots as netplots, then the description of the netleague is missing. of net league table and references “Thus in a ranking table, Treatments were ranked from best to worst along the leading diagonal. Above the leading diagonal were estimates from pairwise meta-analyses, below the leading diagonal were estimates from network meta-analyses.”
Refs: http://dx.doi.org/10.1016/j.ctcp.2020.101260 ; https://doi.org/10.1016/j.rehab.2021.101602

With regard to the display of the results especially of tables, the footnotes describing the table itself are completely missing. They must be settling components of the manuscript. Avoid alphabetic abbreviations, other abbreviations. They should be made explicit not made even more cryptic

Round 2

Reviewer 3 Report

Dear Authors,

The manuscript is methodologically more rigorous, I can only suggest removing the future work paragraph. On the fact that there is a lot to do I do not know what insight leads to the paper, moreover by convention the limitations give rise to conclusions.

I leave just a few imperfections regarding the abstract

Insert registration code in the abstract

16 describe PSA acronym

20 RoB-2
